# Lionfish Diet Composition at Three Study Sites in the Aegean Sea: An Invasive Generalist?

**Ioannis E. Batjakas** [1,*], **Athanasios Evangelopoulos** [2], **Maria Giannou** [1], **Sofia Pappou** [1], **Eleftheria Papanikola** [1], **Maria Atsikvasi** [1], **Dimitris Poursanidis** [3] **and Chrysoula Gubili** [2,*]

1   Department of Marine Sciences, University of Aegean, University Hill, Lesvos Island, 81100 Mytilene, Greece;
    mgiannou1999@gmail.com (M.G.); mard18005@marine.aegean.gr (S.P.); mar18080@marine.aegean.gr (E.P.);
    mar18008@marine.aegean.gr (M.A.)
2   Hellenic Agricultural Organisation—DIMITRA, Fisheries Research Institute, Nea Peramos,
    64007 Kavala, Greece; a.evangelopoulos@inale.gr
3   Foundation for Research and Technology-Hellas (FORTH), Institute of Applied and Computational
    Mathematics, Remote Sensing Lab, 70013 Heraklion, Greece; dpoursanidis@iasm.forth.gr
*   Correspondence: jbatzakas@aegean.gr (I.E.B.); c.gubili@inale.gr (C.G.)

**Abstract:** The diet of the lionfish (*Pterois miles*), an invasive species in the Aegean Sea, was examined by collecting stomach content data from fish collected in three study sites in the Aegean Sea (southern Crete, Kastellorizo, and Nysiros islands). Prey composition in terms of numerical abundance and frequency of occurrence was used to compare lionfish's diet between these sites. Lionfish largely preyed upon teleosts (4% to 83% numerical abundance and 16% to 58% frequency of occurrence, depending on the site) and decapods (12% to 95% numerical abundance and 11% to 81% frequency of occurrence). The most important teleost families in lionfish's diet were Gobiidae, Labridae, and Scorpaenidae, while decapods and especially the family Scyllaridae and the genus *Plesionika* were the dominant decapod prey items. The lionfish was found to be an especially successful generalist across the study sites, an opportunistic, predatory species overall, and at the same time, at a local level, it seems to be an equally successful specialist that could increase the predation mortality of already stressed prey populations and can be a serious threat to endemic, critically endangered, and/or commercially important species.

**Keywords:** *Pterois miles*; lionfish; diet; Gobiidae; Scorpaenidae; Scyllaridae; invasive species; Aegean Sea

**Key Contribution:** This study provides first-time insights into lionfish diet composition in three study sites in the Aegean Sea and highlights the specialist behavior of an especially successful generalist at a local level.

## 1. Introduction

The Mediterranean Sea has become a hotspot for alien species, with an increase in established taxa of 40% in the last decade, with approximately 1000 non-indigenous species being recorded till the end of 2021 [1]. Amongst them, fish (in total 173 species) is the group that attracted the highest attention as their settlement has raised serious concerns due to their rapid range expansion [2,3]. The successful establishment into their newly invaded ecosystems could be attributed to the multiple vectors of introduction such as increased marine traffic, enlargement of the Suez Canal, the shifts in abiotic factors (e.g., habitat quality and climate) [4], and their generalist nature [5,6], which has fundamental effects on local food web dynamics [7,8]. The invasive lionfish *Pterois miles* (Bennett 1828) is one of the most successful invaders [9], with increased predation rates on native fauna, resulting in altered community structure [10–16]. Its presence can reduce the recruitment of native species, drives declines in populations [14,17], and subsequently has serious implications on marine ecosystem functioning [18].

Since the first documented appearance of *Pterois miles* in the Mediterranean Sea in 1991 [19], the species has been extremely successful in establishing populations in new marine ecosystems [20]. Its range is constantly updated in the Mediterranean Sea [21,22], confirming its successful introduction and progressive invasion of the basin. Currently, it is established in the Levantine Sea, in the southern and central Aegean Sea, and in the Greek Ionian Sea, whereas few individuals have been recorded from Tunisia and southern Sicily (Italy) [22]. It reached the Mediterranean Sea through the Suez Cana [19]; however, the introduction pathway to the western Atlantic remains unknown [23]. It was first recorded off Florida in 1985 [12], with exponentially growing populations along the newly colonizing areas of the western Atlantic and the Caribbean region (more than 7.3 million km$^2$). Moreover, a combination of the biological characteristics of the species, such as early maturation and reproduction [12], promotes its range expansion, which has not been interrupted by eradication programs [21]. Its population dramatic increases could be also attributed to its predatory behavior, whereas both native predators and prey are not prepared to face the versatile ecology of the species as seen in both the Mediterranean Sea and the western Atlantic Ocean [12]. Particularly, lionfish diet composition has exhibited a large variability among different locations [24], rendering important location-based diet assessments to better inform local management regimes.

The species exhibits an opportunistic, generalist feeding behavior, whose diet habits are directly connected to prey availability [25,26]. Differences in diet have been reported in the Mediterranean basin, where sampling (spear gun, boat-seining, long lines, and video recordings) and identification approaches (macroscopic examination and visual-video records) revealed that various fish species were among its main prey in Rhodes Island [27], whereas fish or benthic invertebrates were found in stomachs from Cyprus [28,29]. Given that regional differences in its diet are already confirmed, identification of new prey species should be expected with the investigation of its trophic preferences across its invaded geographic range. Therefore, new studies are required to evaluate lionfish diet habits and their effects as a predator of the native fauna. This study aims to provide first-time insights into lionfish diet composition in three study sites in the Aegean Sea (southern Crete, Kastellorizo, and Nysiros islands) and verify the species' generalist strategy as a consumer across sites and individuals.

## 2. Materials and Methods

### 2.1. Ethics Statement

No ethical approval was required for fish provided by local fishermen dead.

### 2.2. Sample Collection

Individuals of *P. miles* were collected between November 2021 to September 2022 from three areas in Greece (southern Crete, Kastellorizo, and Nisyros Islands; Figure 1). All fish were caught as bycatch on nets by local fishermen at depths ranging from 10–20 m. Samples were preserved at −20 °C until further processing. Specimens were measured in length (TL) and weighed to the nearest 0.1 cm and 0.01 g, respectively. The sex of each specimen was determined, and individuals were grouped into three categories (female, male, and unknown). Individuals were also grouped into two size classes, small and large. TL of 17.5 cm was arbitrarily chosen as a threshold value for the separation of the size classes. This TL value equals the length at maturity (L50) for *P. miles* females estimated by Morris [30] based on pooled samples from worldwide locations.

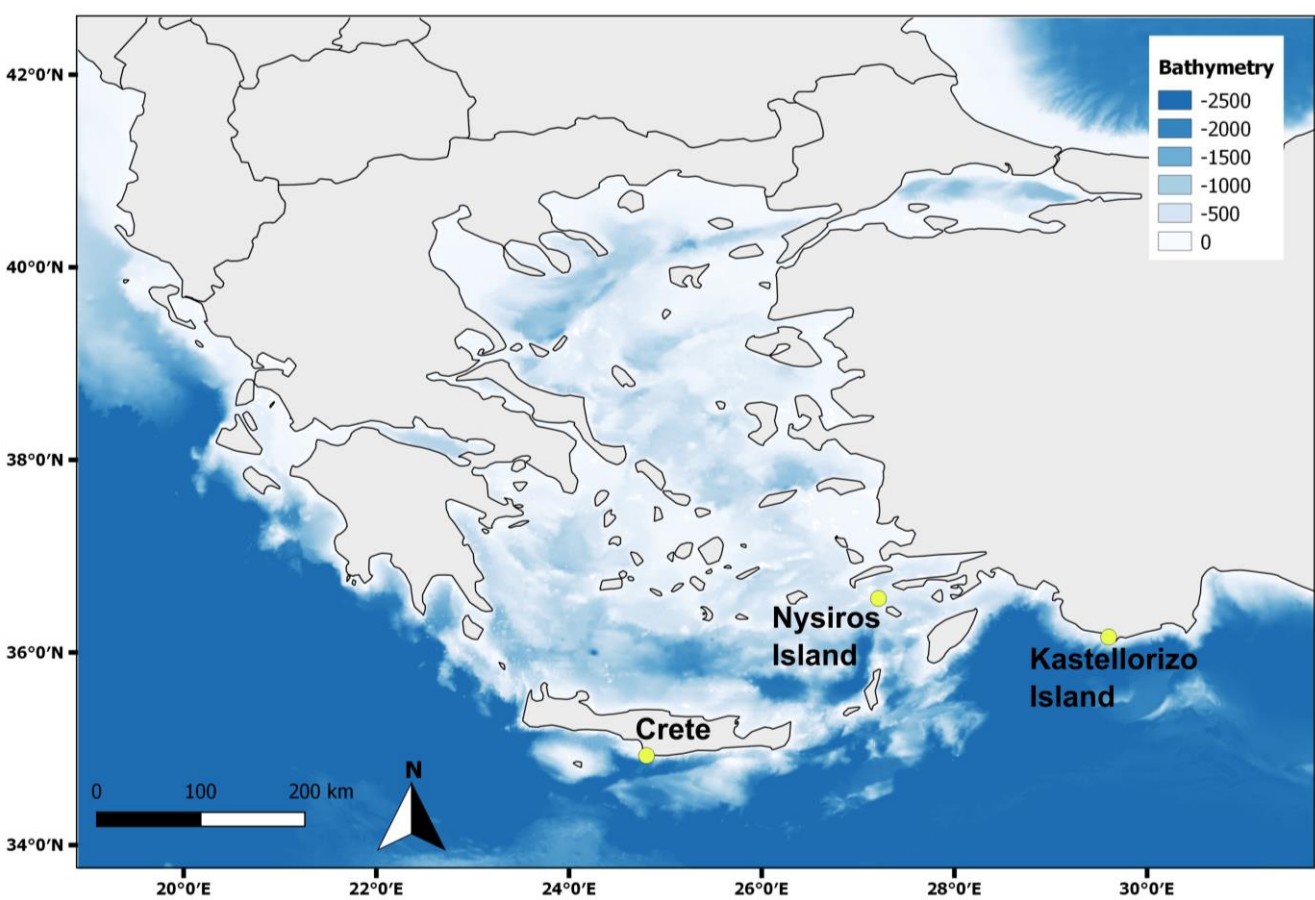

**Figure 1.** Approximate sampling locations of *Pterois miles* in southern Greece.

### 2.3. Lab Work

Each individual was dissected, and its stomach was excised, weighed and its state (empty or non-empty) was determined. The contents of non-empty (="full") stomachs were removed, weighed with an OHAUS Adventurer precision scale and visually examined in a Petri dish under a stereomicroscope (Olympus SZ65). Prey items were identified to the lowest possible taxonomic level and counted. Hard diagnostic parts (i.e., fish bones, otoliths, shrimp rostra, and molluscan shells) were used for taxa identification. Otolith species identification was based on the otolith atlas [31]. Prey remains of the same taxonomic group were grouped together. Stomachs with unidentifiable material (because of advanced digestion) were excluded from further analysis regarding prey items but were not considered empty. Prey taxa were classified into three broad groups: fish, decapods, and benthic invertebrates (including benthic crustacean taxa except decapods).

### 2.4. Data Analysis

To evaluate whether the number of fish stomachs examined was adequate for a valid description of the species' diet, prey accumulation curves [32,33] were computed with the vegan R package [34] for the whole dataset and each study area. The estimated (mean) number of prey groups and associated 95% confidence intervals were plotted against the cumulative number of stomachs examined. Stomach order was randomized as suggested by Ferry and Cailliet [32]. Proportions of empty (vacuity index, VI) and full stomachs were estimated as a percentage of the total number of examined stomachs for each area, sex, and size class. The proportions of empty and full stomachs were tested for significant differences between areas, sexes, and size classes using Pearson's $\chi2$ test of independence. Feeding intensity was also estimated with the ratio of (wet) food weight to total body

weight (in 0/00) (repletion index, RI). RI values were tested for significant differences between areas, sexes, and size classes by means of the Kruskal–Wallis test.

The contribution of each prey taxon i in *P. miles* diet was estimated with the following methods [35]:

(i)     Frequency of occurrence:

$$\%F = Si \times 100/Sf$$

where %F is the frequency of occurrence of prey taxon i in the analyzed stomachs, Si is the number of stomachs in the analysis containing items of prey taxon i, and Sf is the total number of stomachs in the analysis.

(ii)     Numerical:

$$\%N = ni \times 100/\Sigma ni$$

where %N is the relative numerical abundance of prey taxon i, ni is the total number of prey i items, and Σni the total number of all prey items in all stomachs in the analysis.

Visualization of the variations of the relative numerical abundances of prey taxa and groups between areas was carried out using the treemap R package [36].

Non-metric Multi-Dimensional Scaling, nMDS [37], was used to ordinate samples on a 2D plot for the visualization and exploration of the Bray–Curtis similarity matrix, which was calculated based on square root-transformed prey numerical abundance data across all the analyzed stomachs. Permutational multivariate analysis of variance, PERMANOVA [38], was run on the same similarity matrix to test for statistically significant differences in stomach contents composition between areas, sexes, and size classes. All factors were set in the analysis design as fixed, the sums of squares type selected was Type III (partial), the permutation method was a permutation of residuals under a reduced model, and the number of permutations selected was 9999.

The one-way similarity percentage analysis, SIMPER [39], was also run on the similarity matrix to detect the prey taxa responsible for the between-areas dissimilarities (discriminating taxa) and within-area similarities (typifying taxa) regarding the *P. miles* stomach contents prey composition.

Diet overlap by area, sex, and size class was estimated with the Schoener index, Cxy [40]:

$$Cxy = 1 - 0.5 \times \left( \sum |pxi - pyi| \right)$$

where pxi and pyi are the proportions of prey category i (in terms of numerical abundance) in the diet of the species in the area, sex, or class size x and y, respectively. Cxy ranges from 1 (same prey items in the same proportions) to 0 (no common prey items).

The species feeding strategy was graphically depicted using a 2-D representation, where the prey-specific abundance of prey taxon i (Pi) was plotted against its frequency of occurrence (%F) in the stomachs with food contents. This method is a modified Costello graphical analysis [41], and it assesses simultaneously the prey importance, the feeding strategy, and the inter- and intra-individual components of trophic niche width. This information is obtained by the examination of the distribution of the points representing prey categories in the produced plots across the bottom left-top right diagonal (rare vs. dominant prey categories), top-bottom axis (specialization vs. generalization in the diet), and top left-top right axis (specialization at the individual vs. at the population level).

Diet breadth was calculated for each area, sex, and size class using the standardized Levins [42] niche breadth measure [43]:

$$BA = (\Sigma pi^2 - 1)/(N - 1)$$

where pi is the relative abundance of prey taxon i, and N is the total number of prey taxa. The values that this index may take range between 0 and 1, with low values indicating a specialist predator and high values a generalist one. Prey taxa with relative abundance values < 3% and unidentifiable remains were excluded from the analysis.

All analyses were performed using the R Statistical Software (v4.2.2; R Core Team 2022) [44], except the NMDS and SIMPER, which were implemented in PRIMER 6.1.18 [45,46] and the PERMANOVA test, which was carried out in PERMANOVA 1.0.8 [47].

## 3. Results

### 3.1. Sample Size Adequacy

A total of 141 *P. miles* individuals were collected from the three areas (Table 1). More than half of them (73) were collected in Crete, whereas similar numbers were gathered from Kastellorizo (31) and Nisyros (37) islands. Most of the individuals were females (55%), while it was not possible to determine the sex of several fish (34%) due to the early developmental stage of the gonads. The two size classes were comparable in the numbers of individuals (S = 68, L = 73). The prey accumulation curves that were computed for each area (Figure 2) revealed upon visual examination that the numbers of stomachs collected were sufficient for Nisyros, less so for Kastellorizo, whereas for Crete the stomachs sample size was apparently not adequate. However, the estimated uncertainty was high in the cases of Nisyros and Kastellorizo.

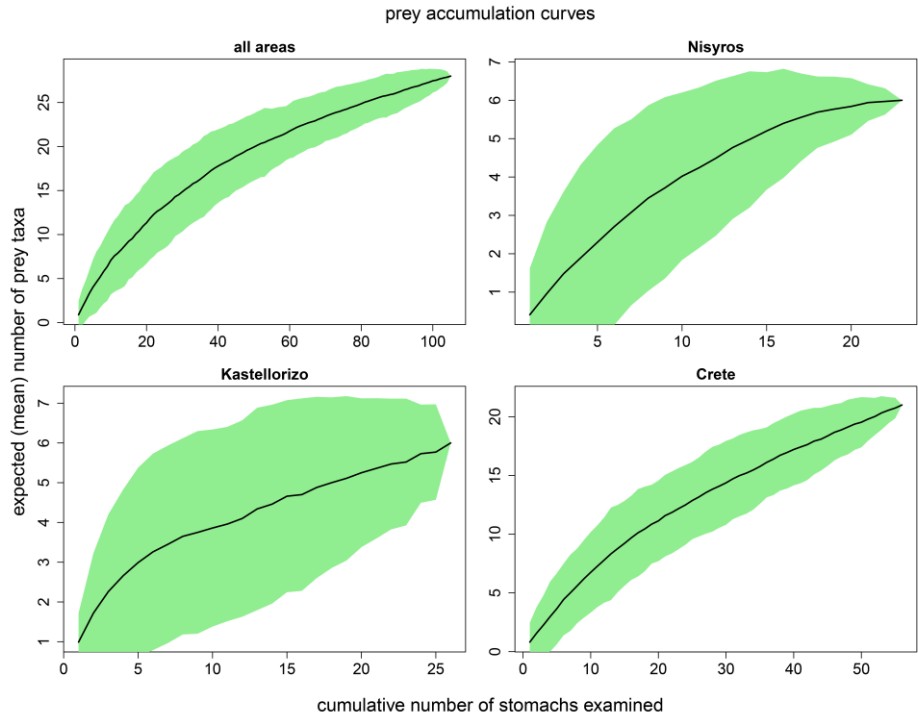

**Figure 2.** Plots of the expected (mean) number of prey taxa as a function of the cumulative number of P. miles stomachs examined (prey accumulation curves) for the whole dataset and separately for Nisyros, Kastellorizo, and Crete. 95% confidence intervals for the estimate are indicated in green.

**Table 1.** Total number of stomachs and percentages of full and empty stomachs (=VI) of *P. miles* for each area, sex (Female, Male, and Unknown), and size class (Small, Large).

| Factor | Levels | Total | Full % | Empty % (=VI) |
|---|---|---|---|---|
| Area | Crete | 73 | 77% | 23% |
| | Kastellorizo | 31 | 84% | 16% |
| | Nisyros | 37 | 62% | 38% |
| Sex | F | 78 | 78% | 22% |
| | M | 15 | 60% | 40% |
| | U | 48 | 73% | 27% |
| Size | S | 68 | 76% | 24% |
| | L | 73 | 73% | 27% |
| Grand Total | | 141 | 74% | 26% |

### 3.2. Pterois miles Feeding Intensity

Overall, 105 of the collected stomachs had prey items inside, and 36 stomachs were empty (Table 1). Empty stomachs were more numerous in individuals from Nisyros and in males, whereas their numbers were similar between small and large individuals. However, the results of the Pearson's χ2 test did not reveal significant differences in the VI values between areas, sexes, or size classes at a significance level of 0.05. The Kruskal–Wallis test revealed that only the area had a significant effect on RI ($\chi2 = 29.561$, $p = 3.809 \times 10^{-7}$). The highest mean values of the repletion index were calculated in individuals from Kastellorizo and the lowest in individuals from Nisyros (Table 2).

**Table 2.** Mean values of the *P. miles* repletion index (0/00) for each area, sex (Female, Male, and Unknown), and size class (Small, Large).

| Sex | Size | Nisyros | Kastellorizo | Crete |
|---|---|---|---|---|
| F | | 31.83 | 315.07 | 171.12 |
| | S | 27.66 | 294.77 | 210.83 |
| | L | 32.66 | 330.3 | 103.39 |
| M | | 27.19 | 187.33 | |
| | S | | 126.32 | |
| | L | 27.19 | 614.39 | |
| U | | 96.94 | 230.66 | 83.19 |
| | S | 23.5 | 166.26 | 100.78 |
| | L | 121.42 | 311.17 | 61.2 |
| | Area | 52.07 | 257.6 | 138.6 |

### 3.3. Contribution of Prey Taxa to P. miles Diet

Overall, the contributions of fish and decapods were comparable in the diet of the species in the study area (%F = 47 and 37, respectively) and much higher than that of benthic invertebrates (%F = 7, Table 3, Figure 3).

**Table 3.** Frequency of occurrence (%F) values for the different *P. miles* prey taxa and groups for each area (Nisyros, Crete, Kastellorizo), sex (Male, Female, Unknown), size class (Small, Large), and for the whole dataset.

| Taxon | N | C | K | F | M | U | S | L | ALL |
|---|---|---|---|---|---|---|---|---|---|
| **BENTHIC INVERTEBRATES** | **5.41** | **9.59** | **3.23** | **8.97** | **6.67** | **4.17** | **5.88** | **8.22** | **7.09** |
| Cumacea | 5.41 | 0.00 | 0.00 | 0.00 | 6.67 | 2.08 | 0.00 | 2.74 | 1.42 |
| Isopoda | 0.00 | 1.37 | 0.00 | 1.28 | 0.00 | 0.00 | 1.47 | 0.00 | 0.71 |
| Gastropoda | 0.00 | 1.37 | 0.00 | 1.28 | 0.00 | 0.00 | 0.00 | 1.37 | 0.71 |
| Mollusca | 0.00 | 1.37 | 0.00 | 1.28 | 0.00 | 0.00 | 0.00 | 1.37 | 0.71 |
| Polychaeta | 0.00 | 4.11 | 0.00 | 3.85 | 0.00 | 0.00 | 2.94 | 1.37 | 2.13 |
| Ostracoda | 0.00 | 1.37 | 3.23 | 1.28 | 0.00 | 2.08 | 1.47 | 1.37 | 1.42 |
| **DECAPODS** | 10.81 | 31.51 | 80.65 | 38.46 | 33.33 | 35.42 | 44.12 | 30.14 | 36.88 |
| **Brachyura** | | | | | | | | | |
| *Homola barbata* (Fabricius, 1793) | 0.00 | 5.48 | 0.00 | 2.56 | 0.00 | 4.17 | 4.41 | 1.37 | 2.84 |
| *Inachus* sp. | 0.00 | 2.74 | 0.00 | 1.28 | 0.00 | 2.08 | 2.94 | 0.00 | 1.42 |
| **Natantia** | | | | | | | | | |
| *Alpheus* sp. | 0.00 | 0.00 | 3.23 | 1.28 | 0.00 | 0.00 | 0.00 | 1.37 | 0.71 |
| *Plesionika edwardsii* (Brandt, 1851) | 0.00 | 0.00 | 41.94 | 8.97 | 20.00 | 6.25 | 10.29 | 8.22 | 9.22 |
| *Plesionika* spp. | 0.00 | 1.37 | 29.03 | 6.41 | 6.67 | 8.33 | 5.88 | 8.22 | 7.09 |
| Caridea | 8.11 | 1.37 | 0.00 | 3.85 | 0.00 | 2.08 | 5.88 | 0.00 | 2.84 |
| *Parapenaeus longirostris* (Lucas, 1846) | 0.00 | 2.74 | 0.00 | 0.00 | 0.00 | 4.17 | 2.94 | 0.00 | 1.42 |
| Natantia | 0.00 | 0.00 | 12.90 | 2.56 | 6.67 | 2.08 | 4.41 | 1.37 | 2.84 |
| **Macrura** | | | | | | | | | |
| *Scyllarides latus* (Latreille, 1803) | 0.00 | 6.85 | 0.00 | 3.85 | 0.00 | 4.17 | 2.94 | 4.11 | 3.55 |
| *Scyllarus arctus* (Linnaeus, 1758) | 0.00 | 1.37 | 0.00 | 1.28 | 0.00 | 0.00 | 1.47 | 0.00 | 0.71 |
| *Scyllarus* sp. | 0.00 | 9.59 | 0.00 | 6.41 | 0.00 | 4.17 | 5.88 | 4.11 | 4.96 |
| Scyllaridae larvae | 0.00 | 1.37 | 0.00 | 1.28 | 0.00 | 0.00 | 0.00 | 1.37 | 0.71 |
| Scyllaridae | 0.00 | 9.59 | 3.23 | 7.69 | 0.00 | 4.17 | 7.35 | 4.11 | 5.67 |
| Decapoda | 2.70 | 0.00 | 0.00 | 1.28 | 0.00 | 0.00 | 0.00 | 1.37 | 0.71 |
| **FISH** | 51.35 | 57.53 | 16.13 | 52.56 | 20.00 | 45.83 | 45.59 | 47.95 | 46.81 |
| *Atherina hepsetus* Linnaeus, 1758 | 0.00 | 1.37 | 0.00 | 1.28 | 0.00 | 0.00 | 1.47 | 0.00 | 0.71 |
| *Chromis* sp. | 0.00 | 2.74 | 0.00 | 1.28 | 0.00 | 2.08 | 1.47 | 1.37 | 1.42 |
| Gobidae | 5.41 | 2.74 | 0.00 | 5.13 | 0.00 | 0.00 | 2.94 | 2.74 | 2.84 |
| Labridae | 5.41 | 0.00 | 0.00 | 1.28 | 6.67 | 0.00 | 0.00 | 2.74 | 1.42 |
| *Pterois* sp. | 5.41 | 0.00 | 0.00 | 2.56 | 0.00 | 0.00 | 0.00 | 2.74 | 1.42 |
| *Sargocentron rubrum* (Forsskål, 1775) | 0.00 | 1.37 | 0.00 | 1.28 | 0.00 | 0.00 | 0.00 | 1.37 | 0.71 |
| *Scorpaena scrofa* (Linnaeus,1758) | 0.00 | 1.37 | 0.00 | 1.28 | 0.00 | 0.00 | 0.00 | 1.37 | 0.71 |
| *Spicara smaris* (Linnaeus, 1758) | 0.00 | 1.37 | 0.00 | 0.00 | 0.00 | 2.08 | 0.00 | 1.37 | 0.71 |
| fish remains | 40.54 | 47.95 | 16.13 | 42.31 | 13.33 | 41.67 | 39.71 | 38.36 | 39.01 |

Fish remains, *Plesionika edwardsii*, *Plesionika* spp., Scyllaridae, and *Scyllarus* sp., were the prey items most frequently found in the stomach contents of *P. miles* (%F $\geq$ 5). In terms of relative numerical abundance, the contribution of crustaceans (%N = 61) was higher than that of fish (%N = 35), whereas the relative numerical abundance of benthic invertebrates was small (%N = 4, Table 4, Figure 4). Fish remains, *Plesionika* spp. and *Plesionika edwardsii*, were numerically the most abundant prey items in the stomach contents of the species (%N $\geq$ 5).

**Table 4.** Relative numerical abundance (%N) values for the different *P. miles* prey taxa and groups for each area (Nisyros, Crete, and Kastellorizo), sex (Male, Female, Unknown), size class (Small, Large), and for the whole dataset.

| Taxon | N | C | K | F | M | U | S | L | ALL |
|---|---|---|---|---|---|---|---|---|---|
| **BENTHIC INVERTEBRATES** | **4.88** | **7.89** | **0.71** | **4.55** | **3.45** | **3.33** | **4.12** | **4.00** | **4.07** |
| Cumacea | 4.88 | 0.00 | 0.00 | 0.00 | 3.45 | 1.11 | 1.18 | 0.00 | 0.68 |
| Isopoda | 0.00 | 0.88 | 0.00 | 0.57 | 0.00 | 0.00 | 0.00 | 0.80 | 0.34 |
| Gastropoda | 0.00 | 0.88 | 0.00 | 0.57 | 0.00 | 0.00 | 0.59 | 0.00 | 0.34 |
| Mollusca | 0.00 | 0.88 | 0.00 | 0.57 | 0.00 | 0.00 | 0.59 | 0.00 | 0.34 |
| Polychaeta | 0.00 | 3.51 | 0.00 | 2.27 | 0.00 | 0.00 | 0.59 | 2.40 | 1.36 |
| Ostracoda | 0.00 | 1.75 | 0.71 | 0.57 | 0.00 | 2.22 | 1.18 | 0.80 | 1.02 |
| **DECAPODS** | **12.20** | **36.84** | **95.00** | **60.23** | **65.52** | **61.11** | **62.35** | **59.20** | **61.02** |
| **Brachyura** | | | | | | | | | |
| *Homola barbata* (Fabricius, 1793) | 0.00 | 4.39 | 0.00 | 1.14 | 0.00 | 3.33 | 0.59 | 3.20 | 1.69 |
| *Inachus* sp. | 0.00 | 1.75 | 0.00 | 0.57 | 0.00 | 1.11 | 0.00 | 1.60 | 0.68 |
| **Natantia** | | | | | | | | | |
| *Alpheus* sp. | 0.00 | 0.00 | 1.43 | 1.14 | 0.00 | 0.00 | 1.18 | 0.00 | 0.68 |
| *Plesionika edwardsii* (Brandt, 1851) | 0.00 | 0.00 | 35.00 | 14.77 | 31.03 | 15.56 | 18.82 | 13.60 | 16.61 |
| *Plesionika* spp. | 0.00 | 1.75 | 47.86 | 22.16 | 31.03 | 23.33 | 28.82 | 16.00 | 23.39 |
| Caridea | 9.76 | 0.88 | 0.00 | 2.27 | 0.00 | 1.11 | 0.00 | 4.00 | 1.69 |
| *Parapenaeus longirostris* (Lucas, 1846) | 0.00 | 1.75 | 0.00 | 0.00 | 0.00 | 2.22 | 0.00 | 1.60 | 0.68 |
| Natantia | 0.00 | 0.00 | 10.00 | 6.25 | 3.45 | 2.22 | 4.71 | 4.80 | 4.75 |
| **Macrura** | | | | | | | | | |
| *Scyllarides latus* (Latreille, 1803) | 0.00 | 7.89 | 0.00 | 2.84 | 0.00 | 4.44 | 2.94 | 3.20 | 3.05 |
| *Scyllarus arctus* (Linnaeus, 1758) | 0.00 | 0.88 | 0.00 | 0.57 | 0.00 | 0.00 | 0.00 | 0.80 | 0.34 |
| *Scyllarus* sp. | 0.00 | 7.02 | 0.00 | 3.41 | 0.00 | 2.22 | 1.76 | 4.00 | 2.71 |
| Scyllaridae larvae | 0.00 | 1.75 | 0.00 | 1.14 | 0.00 | 0.00 | 1.18 | 0.00 | 0.68 |
| Scyllaridae | 0.00 | 8.77 | 0.71 | 3.41 | 0.00 | 5.56 | 1.76 | 6.40 | 3.73 |
| Decapoda | 2.44 | 0.00 | 0.00 | 0.57 | 0.00 | 0.00 | 0.59 | 0.00 | 0.34 |
| **FISH** | **82.93** | **55.26** | **4.29** | **35.23** | **31.03** | **35.56** | **33.53** | **36.80** | **34.92** |
| *Atherina hepsetus* Linnaeus, 1758 | 0.00 | 0.88 | 0.00 | 0.57 | 0.00 | 0.00 | 0.00 | 0.80 | 0.34 |
| *Chromis* sp. | 0.00 | 2.63 | 0.00 | 1.14 | 0.00 | 1.11 | 0.59 | 1.60 | 1.02 |
| Gobidae | 7.32 | 1.75 | 0.00 | 2.84 | 0.00 | 0.00 | 1.18 | 2.40 | 1.69 |
| Labridae | 7.32 | 0.00 | 0.00 | 1.14 | 3.45 | 0.00 | 1.76 | 0.00 | 1.02 |
| *Pterois* sp. | 4.88 | 0.00 | 0.00 | 1.14 | 0.00 | 0.00 | 1.18 | 0.00 | 0.68 |
| *Sargocentron rubrum* (Forsskål, 1775) | 0.00 | 4.39 | 0.00 | 2.84 | 0.00 | 0.00 | 2.94 | 0.00 | 1.69 |
| *Scorpaena scrofa* Linnaeus, 1758 | 0.00 | 0.88 | 0.00 | 0.57 | 0.00 | 0.00 | 0.59 | 0.00 | 0.34 |
| *Spicara smaris* (Linnaeus, 1758) | 0.00 | 0.88 | 0.00 | 0.00 | 0.00 | 1.11 | 0.59 | 0.00 | 0.34 |
| fish remains | 63.41 | 43.86 | 4.29 | 25.00 | 27.59 | 33.33 | 24.71 | 32.00 | 27.80 |

### 3.4. Multivariate Analysis of P. miles Diet Composition

Stomach samples from a particular area were, in most cases, clustered together on the nMDS ordination plot, implying differences between areas in *P. miles* diet composition (Figure 5). No clear separation between groups was discerned on the nMDS plot according to sex or size class. The PERMANOVA main test revealed that the diet composition of the species differed significantly between areas [Pseudo-F = 5.0894, *p* (perm) = 0.0001]. Moreover, the PERMANOVA pairwise tests showed that *P. miles* stomach contents differed significantly in composition between Kastellorizo and Nisyros and between Kastellorizo and Crete (t = 2.1535, *p* (perm) = 0.0005 and t = 3.2464, *p* (perm) = 0.0001, respectively). However, the difference in *P. miles* diet composition between Nisyros and Crete was marginally insignificant (PERMANOVA: t = 1.4544, *p* (perm) = 0.0506). No statistically significant differences in the trophic preferences of the species between sexes or size classes were found.

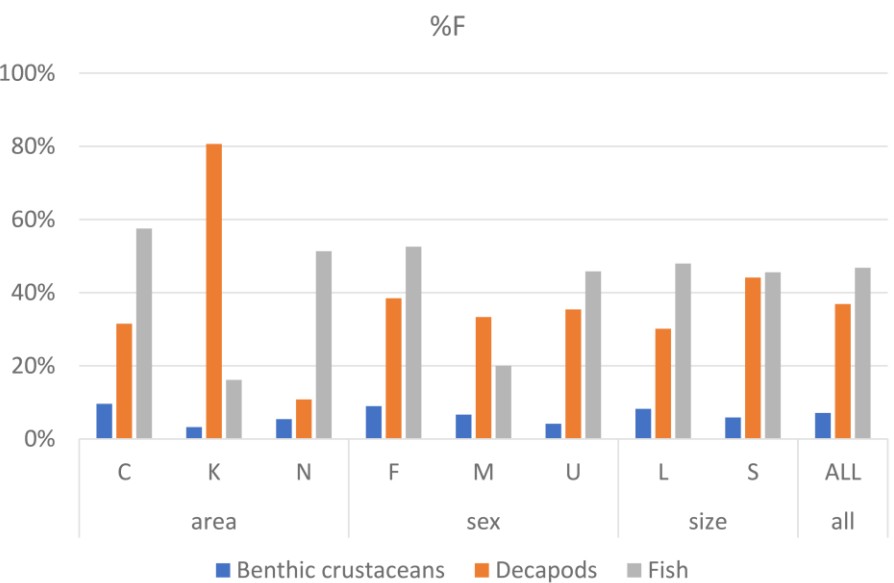

**Figure 3.** Variations of the frequency of occurrence (%F) of the *P. miles* prey groups between areas (Nisyros, Crete, and Kastellorizo), sex (Male, Female, and Unknown), size class (Small, Large), and for the whole dataset.

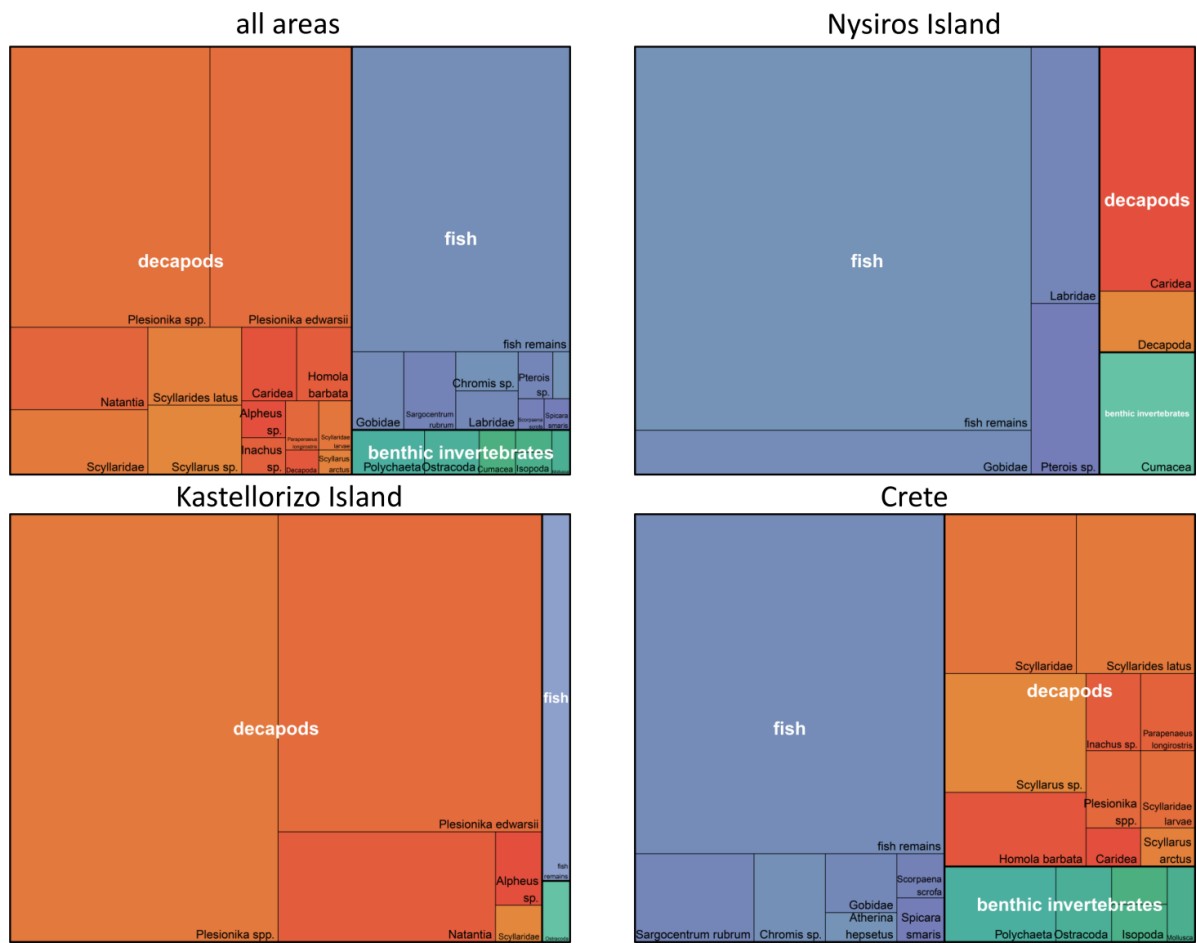

**Figure 4.** Treemap plots depicting the variations of the relative numerical abundance (%N) of the *P. miles* prey groups between areas.

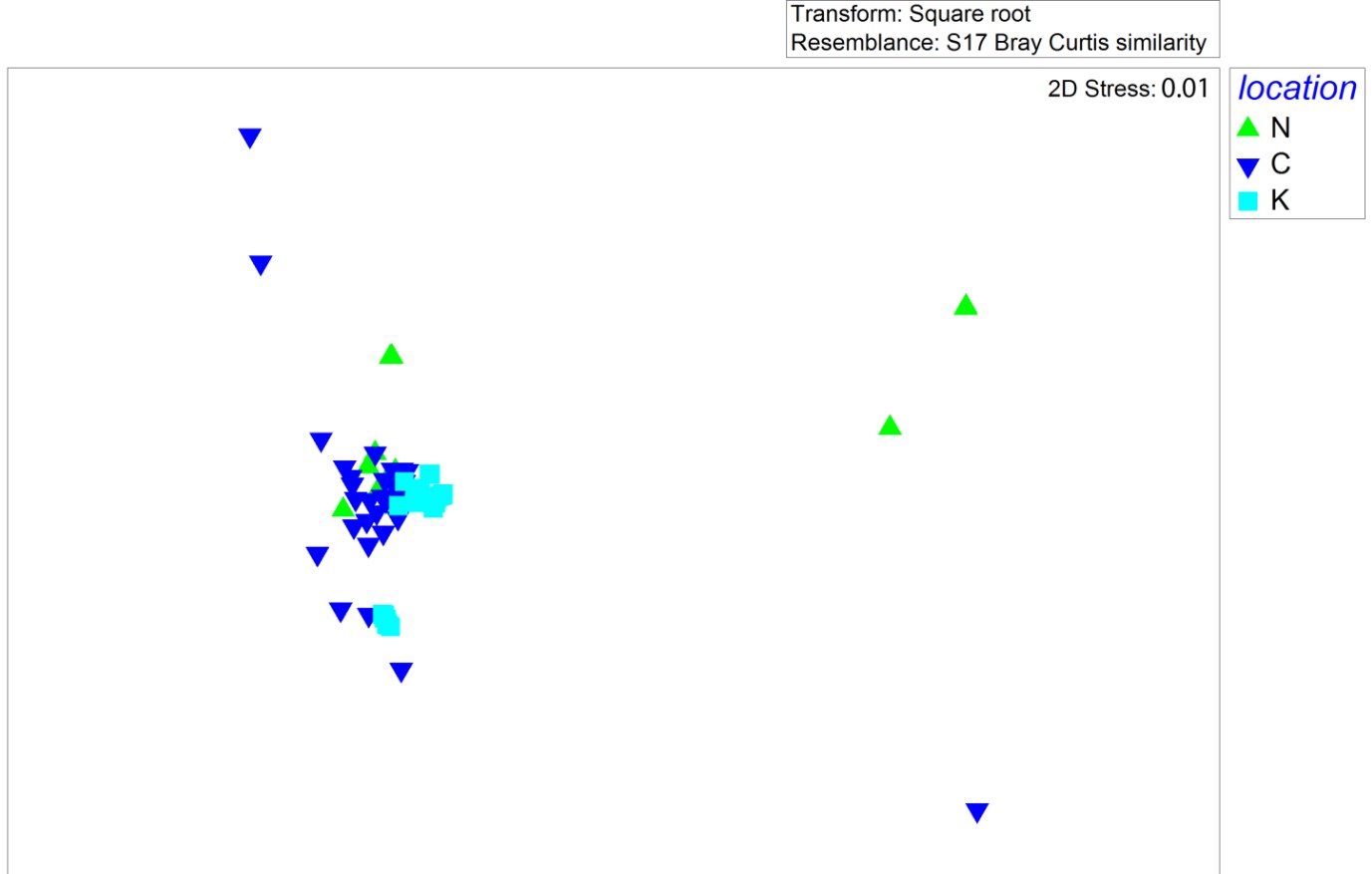

**Figure 5.** nMDS ordination plot of the *P. miles* stomach samples, colored by area (N = Nisyros; K = Kastellorizo; C = Crete).

According to the one-way SIMPER analysis results (Table 5), fish remains were the trophic item that was characteristic of the stomach samples from Nisyros and Crete, whereas, for Kastellorizo, the typifying trophic items were the crustacean taxa *Plesionika edwardsii* and *Plesionika* spp. Fish remains were the main discriminating prey item responsible for the Nisyros and Crete samples in terms of diet composition (contributing 37% of their dissimilarity), while several other prey taxa also contributed to the dissimilarity of the two areas, albeit to a lesser degree. *Plesionika edwardsii*, *Plesionika* spp., and fish remains cumulatively contributed 75.43% and 69.44% of the dissimilarity in the diet composition of *P. miles* between Nisyros and Kastellorizo and Crete and Kastellorizo, respectively.

### 3.5. Pterois miles Diet Overlap between Areas, Sexes, and Size Classes

*Pterois miles* diet overlap was moderate between Nisyros and Crete (Cxy = 0.46) and very low between Kastellorizo and Nisyros or Crete (Cxy = 0.04 and 0.07, respectively). Moreover, diet overlap was considerable between the sexes (Cxy = 0.67) and between size classes (Cxy = 0.69).

### 3.6. Pterois miles Feeding Strategy

The modified Costello method results (Figure 6) indicated that across all areas, decapods and fish were more important prey categories than benthic invertebrates. A certain degree of specialization of *P. miles* in decapods and fish was also identified. Fish was the dominant prey category in the samples from Nisyros. Specialization in fish at the population level and in benthic invertebrates at the individual level was also indicated in the plot for Nisyros. Fish were more important than decapods in the samples from Crete, and benthic invertebrates were the least important prey category. Moreover, the plot for Crete

revealed a certain degree of specialization in decapods and fish in that area. Decapods were the dominant prey category of *P. miles* in the samples from Kastellorizo. The plot also indicated specialization in decapods at the population level.

**Table 5.** Results of the one-way SIMPER analysis comparing areas in terms of the *P. miles* diet composition. The table presents area typifying species and species contributing most to the between areas dissimilarities up to a 90% cut-off value.

| One-way SIMPER analysis | | | | | |
|---|---|---|---|---|---|
| **Within groups** | | | | | |
| **Species** | **Av.Abund** | **Av.Sim** | **Sim/SD** | **Contrib%** | **Cum.%** |
| *Group N (Average similarity = 34.59)* | | | | | |
| fish remains | 0.80 | 32.91 | 0.79 | 95.14 | 95.14 |
| *Group C (Average similarity = 29.05)* | | | | | |
| fish remains | 0.73 | 27.19 | 0.74 | 93.60 | 93.60 |
| *Group K (Average similarity = 26.60)* | | | | | |
| *Plesionika edwarsii* | 0.91 | 15.56 | 0.53 | 58.51 | 58.51 |
| *Plesionika* spp. | 0.91 | 9.28 | 0.35 | 34.90 | 93.41 |
| **Between groups** | | | | | |
| **Species** | **Av.Abund** | **Av.Abund** | **Av.Diss** | **Diss/SD** | **Contrib%** | **Cum.%** |
| *Groups N and C (Average dissimilarity = 69.17)* | | | | | | |
| | Group N | Group C | | | | |
| fish remains | 0.80 | 0.73 | 23.30 | 1.09 | 33.69 | 33.69 |
| Caridea | 0.15 | 0.02 | 5.05 | 0.39 | 7.30 | 40.98 |
| *Scyllarus* sp. | 0.00 | 0.13 | 4.02 | 0.35 | 5.81 | 46.80 |
| Scyllaridae | 0.00 | 0.14 | 3.89 | 0.37 | 5.63 | 52.42 |
| *Pterois* sp. | 0.09 | 0.00 | 3.53 | 0.30 | 5.10 | 57.52 |
| Cumacea | 0.09 | 0.00 | 3.53 | 0.30 | 5.10 | 62.63 |
| Gobidae | 0.10 | 0.04 | 2.96 | 0.34 | 4.29 | 66.91 |
| *Scyllarides latus* | 0.00 | 0.12 | 2.89 | 0.29 | 4.18 | 71.09 |
| Labridae | 0.10 | 0.00 | 2.79 | 0.29 | 4.04 | 75.13 |
| *Homola barbata* | 0.00 | 0.08 | 2.11 | 0.26 | 3.04 | 78.18 |
| Decapoda | 0.04 | 0.00 | 1.76 | 0.21 | 2.55 | 80.73 |
| Polychaeta | 0.00 | 0.06 | 1.65 | 0.23 | 2.39 | 83.12 |
| *Chromis* sp. | 0.00 | 0.04 | 1.50 | 0.18 | 2.18 | 85.29 |
| *Parapenaeus longirostris* | 0.00 | 0.04 | 1.29 | 0.18 | 1.87 | 87.16 |
| *Inachus* sp. | 0.00 | 0.04 | 1.19 | 0.18 | 1.72 | 88.88 |
| Ostracoda | 0.00 | 0.03 | 0.95 | 0.13 | 1.38 | 90.26 |
| *Groups N and K (Average dissimilarity = 93.62)* | | | | | | |
| | Group N | Group K | | | | |
| *Plesionika edwarsii* | 0.00 | 0.91 | 26.14 | 0.91 | 27.92 | 27.92 |
| *Plesionika* spp. | 0.00 | 0.91 | 23.09 | 0.71 | 24.67 | 52.59 |
| fish remains | 0.80 | 0.21 | 21.39 | 1.12 | 22.84 | 75.43 |
| Natantia | 0.00 | 0.27 | 6.06 | 0.40 | 6.48 | 81.91 |
| Caridea | 0.15 | 0.00 | 3.72 | 0.37 | 3.97 | 85.88 |
| *Pterois* sp. | 0.09 | 0.00 | 2.86 | 0.29 | 3.06 | 88.94 |
| Cumacea | 0.09 | 0.00 | 2.86 | 0.29 | 3.06 | 92.00 |
| *Groups C and K (Average dissimilarity = 93.68)* | | | | | | |
| | Group C | Group K | | | | |
| *Plesionika edwarsii* | 0.00 | 0.91 | 24.18 | 0.90 | 25.81 | 25.81 |
| *Plesionika* spp. | 0.03 | 0.91 | 21.79 | 0.72 | 23.26 | 49.08 |
| fish remains | 0.73 | 0.21 | 19.08 | 1.07 | 20.37 | 69.44 |
| Natantia | 0.00 | 0.27 | 5.63 | 0.40 | 6.00 | 75.45 |
| Scyllaridae | 0.14 | 0.04 | 3.57 | 0.40 | 3.81 | 79.26 |
| *Scyllarus* sp. | 0.13 | 0.00 | 3.14 | 0.35 | 3.35 | 82.61 |
| *Scyllarides latus* | 0.12 | 0.00 | 2.35 | 0.29 | 2.51 | 85.12 |
| *Homola barbata* | 0.08 | 0.00 | 1.68 | 0.26 | 1.79 | 86.91 |
| Polychaeta | 0.06 | 0.00 | 1.33 | 0.23 | 1.42 | 88.33 |
| Ostracoda | 0.03 | 0.04 | 1.27 | 0.21 | 1.36 | 89.69 |
| *Chromis* sp. | 0.04 | 0.00 | 1.16 | 0.18 | 1.24 | 90.93 |

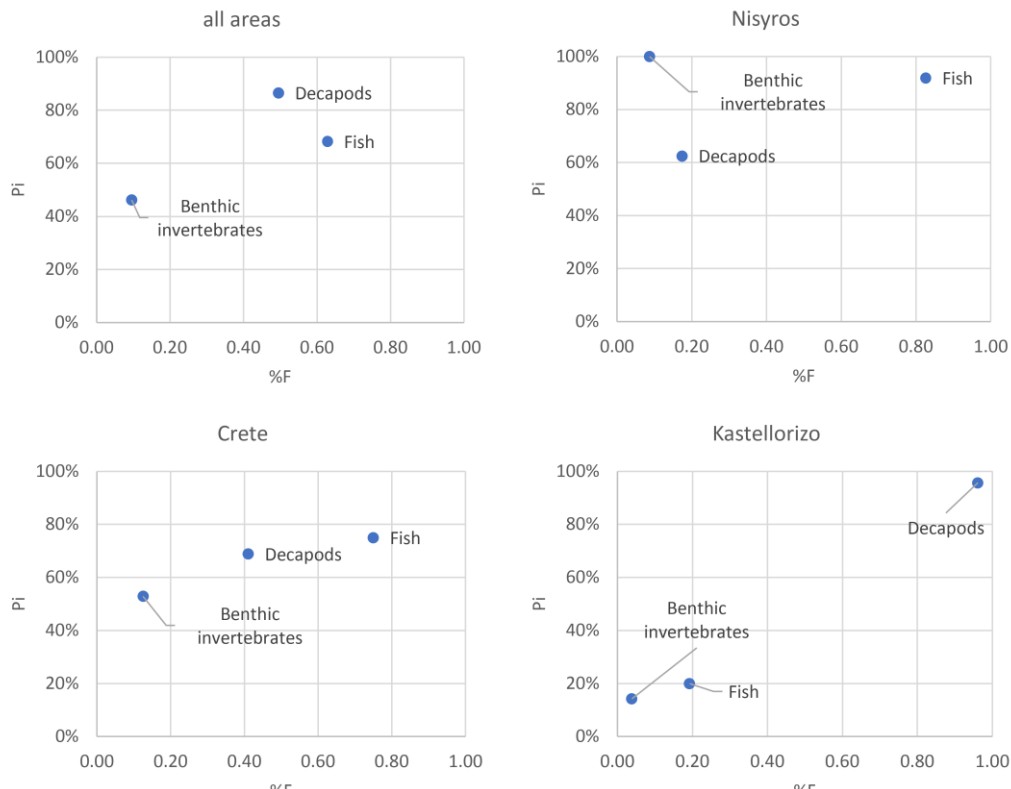

**Figure 6.** Modified Costello graphical analysis of the *P. miles* diet for the whole dataset and separately for each area. The prey-specific abundance of prey category i (Pi) is plotted in the charts against its frequency of occurrence (%F) in the stomachs with food contents.

### 3.7. Pterois miles Diet Breadth

Levins' index values indicated a rather large trophic niche breadth in all areas (BA = 0.68–0.85), with the maximum value of the index calculated for Nisyros. Diet breadth was similar between sexes (BA = 44 and 42 for females and males, respectively) and higher in small (BA = 59) than in large individuals (BA = 44).

### 4. Discussion

The lionfish (*P. miles*) is a scorpaenid fish endemic in the Red Sea and the Gulf of Aqaba, where it preys on a wide variety of benthic fishes and decapods [48,49]. This study provides a first comparative assessment of its diet composition in three different study areas located in southern Greece (Southern Aegean Sea) and highlights the similarities and differences in its feeding habits. To our knowledge, this is the first study in Greece to describe the species' diet composition and feeding patterns at a regional scale and to compare them among different areas.

The diet of *P. miles* was dominated either by fish or by decapods, depending on the area. The contribution of other benthic invertebrate groups in the species' diet was comparatively low across all areas. At the same time, the lionfish diet composition and the relative contributions of prey varied considerably among study areas. Decapods were by far the primary prey in numerical abundance (95%) and frequency of occurrence (80.65%), followed by fish (4.3% and 16.1%, respectively) in Kastellorizo Island. Conversely, the main prey was fish (82.9% numerical abundance and 51.4% frequency of occurrence, respectively), followed by decapods (12.2% numerical abundance and 10.8% frequency of occurrence, respectively) in Nisyros Island. A similar lionfish prey composition was reported by Morris Jr and Akins [50], who stated that 71.8% numerical abundance and 61.6% frequency of occurrence of the prey species of lionfish in the Bahamas were teleosts and crustaceans. The

dominant presence of these taxa as prey items in lionfish's diet was noted in other studies as well [15,24,27,49–53].

*Pterois miles* seems to behave as a specialist predator in both sites, targeting specific food items. Almost all 71%F and 83%N of its diet comprised of *Plesionika* spp. in Kastellorizo Island, whereas on the island of Nisyros, most prey (40.5%F and 63.4%N) were unidentified fish remains, whilst the families Gobidae (5.4%F and 7.3%N) and Labridae (5.4%F and 7.3%N) dominated the identified fish prey (5.4%F and 4.88%N). Additionally, almost all the decapod prey items belonged to caridean shrimp (8.11%F and 9.76%N) in Nisyros Island. Similarly, the diet of *P. miles* was composed predominantly of bony fish (78.5%N), with the majority of prey belonging to the family Gobidae, followed by Pomacentridae and Labridae in Rhodes Island, southeastern Aegean Sea [27]. Fish prey that belongs to the aforementioned families were also reported in the Caribbean Sea, such as the Mexican Caribbean [51,52], Costa Rica [53], and Puerto Rico [15]. These findings support the hypothesis that lionfish can be dietary specialists [18]. Specialization in diet may largely depend on local prey assemblages' composition, and thus, it is more likely to be observed locally [18,54].

Fish and decapods were also the main prey categories for the lionfish (%N = 55.3 and 36.8 and %F = 57.5% and 31.5%, respectively) in southern Crete. In this site, the lionfish exhibited a relatively more balanced diet, with one noticeable exception. Interestingly, a large proportion of the decapod prey belonged to the family Scyllaridae (26.3%N out of 36.8%N and 28.8%F out of 42.5%F). The specialist behavior appeared here as well, but to a lesser degree than in Kastellorizo Island. Thus, the lionfish could pose a threat to the endangered Mediterranean slipper lobster species (Scyllaridae), at least at the local level. Native Mediterranean scorpionfish species may prey on slipper lobsters, but only in one study, to our knowledge; *S. latus* and *S. arctus* were both listed amongst the prey items of *S. scrofa* [55].

It is difficult to properly assess the actual fisheries pressure on threatened and/or protected decapods when relying on official data [56,57]. The degree of uncertainty increases in species with limited or no commercial value, such as the slipper lobsters of the genus *Scyllarus*. The addition of the pressure caused by the lionfish predation, along with the uncertainty level of the fisheries pressure, may further reduce *Scyllarus* populations.

Native Mediterranean fish species of the Scorpanidae family exhibit several ecological similarities with *P. miles*, such as from being a generalist to a specialist strategy at a local level. For instance, *Scorpaena maderensis* Valenciennes, 1833 prefers epibenthic crustaceans [58], and *Scorpaena loppei* Cadenat, 1943 prefers mysids and decapods [59]. Studies regarding the feeding ecology of *S. porcus*, showed similar specialist feeding strategies [60–63], and in some cases, endangered seahorse species were preyed upon [64]. However, in all studies investigating the feeding habits of *P. miles*, it is suggested that many factors, such as prey availability, habitat complexity, and season could affect the feeding ecology of the species.

## 5. Conclusions

In conclusion, the lionfish is an especially successful generalist, opportunistic, predatory species at a regional scale [18,24,50,54,65], and as such, it feeds on the most abundant and common prey species [12]. At the same time, at a local level, it seems to be an equally successful specialist, and it could increase the predation mortality of already stressed prey populations, depending on local predator communities [24]. It can have a high ecological impact on native Mediterranean communities [66], similar to the detrimental impacts on native fish fauna and the food web in the Caribbean ecosystem [12,20,67–69], and can be a serious threat to endemic, critically endangered [17,70,71], and/or commercially important species [52]. Our results indicate that this is the case at our study sites in the Aegean Sea.

However, in order to reveal individual- and population-level specializations in lionfish diet and whether these can cause negative effects on native and/or endangered prey

populations, robust large-scale studies of the species' diet composition in association with prey availability are needed.

**Author Contributions:** Investigation, M.G., S.P., E.P. and M.A.; Resources, D.P.; Formal analysis, Visualization, A.E.; Writing—original draft, I.E.B., A.E. and C.G.; Writing— review & editing, I.E.B., A.E. and C.G.; Supervision, I.E.B.; Funding acquisition, C.G. All authors have read and agreed to the published version of the manuscript.

**Funding:** This work has been financed by the Operational Programme of Fisheries and Sea (OPFS) 2014–2020, Greece, and the European Maritime and Fisheries Fund (EMFF) as part of the project "Monitoring and control of invasive alien species in Greece using innovative techniques under current and future climate conditions—INVASION" (MIS 5049543).

**Institutional Review Board Statement:** Not applicable. No ethical approval was required for fish provided by local fishermen dead.

**Data Availability Statement:** Not applicable.

**Acknowledgments:** The authors are very grateful to two anonymous reviewers for their valuable comments and to local fishermen for providing samples.

**Conflicts of Interest:** The authors declare no conflict of interest.

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
