# Peer review of "Lionfish Diet Composition at Three Study Sites in the Aegean Sea: An Invasive Generalist?"

_fishes, doi:10.3390/fishes8060314_

Round 1

Reviewer 1 Report

(Same comments given to editor.)

REVIEW

Fishes

Batjakas et al. “Lionfish diet composition in in three study sites in the Aegean Sea (southern Crete, and Kastellorizo and Nysiros islands): An invasive generalist?”

General comments: The authors present the results of a descriptive study of lionfish gut contents from samples contributed by fishermen at three sites in the Aegean Sea. A variety of statistical approaches are used to show differences and similarities in stomach content composition among the three sites, and these analyses are used to discuss lionfish feeding behavior with respect to generalist vs. specialist strategies. While I thought the paper was adequate as a simple description of the gut contents differences, I thought the lack of characterization of potential differences in prey availability and habitat type among the three sites weakened the authors’ inferences about dietary generalist vs. specialist behavior. There were also a number of typographical errors and figures and tables that were formatted in difficult to interpret ways and lacking clear explanation in the figure and table legends. A lack of explanation in sections of the methods also led to ambiguity that needs to be resolved. I think the paper could be reconsidered after revision, but is not suitable for publication in its current state.

Line by line comments:

17-19: It’s unclear what sampling unit the low and high ends of the percentage ranges are based on. Are these maximum and minimums per individual fish sampled? Maximum and minimum site means?

21-25: How lionfish can be considered both generalists and specialists, in different context or scales, is a complex idea that is not explained as clearly as it could be here. This concept seems to be a crux of the hypothesis and interpretation of the study, and should be developed better in the introduction, as well.

31: Of the ~1000 documented invasive species, how many are fishes, and what are the primary vectors of invasion?

34: It is unclear what “short migration rate” means. Is this referring to rapid range expansion of invading fish populations?

41: It would be helpful to state the year of the first appearance of lionfish in the Mediterranean and briefly review the invasion history there in comparison with the invasion history in the Western Atlantic. Important insights into how environmental conditions, native prey communities, and management styles affect the invasion could be gained by consideration of the results of this study in relation to results of published studies from the Western Atlantic. In general, aiming for greater, global insights into the lionfish species and invasion ecology in general would improve the paper.

66: It is implied that the fish were provided by local fishermen. Do you have any information on where and how they were collected, i.e. by hook and line, nets, or spearing, and at approximately what depths and in what kinds of habitats they were collected? Were they caught as bycatch or through targeted efforts by the fishermen? As much information as possible about the environments in which the fish were collected should be presented, even if it is qualitative information.

68: 0.01 mm is an exceedingly high precision that seems unrealistic and unnecessary for TL measurements of adult fish. Is this a typographical error?

76: How were the fish preserved until processing- frozen?

109: Was there a wide range in size among prey species such that %F and %N might both exaggerate the importance of small but numerous species in the diet of lionfish? If so is there any way you might calculate a sort of prey weight-adjusted importance index? This potential caveat of interpretation should be discussed, at least.

133-137: It is unclear what aspects of the pi vs %F plot provide information about feeding strategy and trophic niche width. A brief review of how this biplot method works, and what patterns would be expected for different feeding strategies, would be helpful.

164: The legend for Figure 2 needs to explain what the green clouds are. I assume they are some kind of minimum and maximum values from resampling with randomization of stomach order.

180-190: The semi-hierachical classification of species and groups in the stomach contents table is confusing, especially because there is some overlap in categories. E.g., “benthic invertebrates” includes some crustaceans. The table legend includes some typographical errors and non-English letters.

189: Would it be possible to put confidence intervals or other error bars on the %F values in Figure 3? The figure legend also needs a little more description and has some typographical errors.

203: Figure 4 is helpful, but the small, dark colored fonts of the sub-taxon labels are difficult to read.

225: Does each marker in figure 5 represent an individual fish stomach analyzed? It seems like there should be more markers, especially for Nisyros.

311: Figure 6 seems to be missing Kastellorizo.

(Same comments given to authors.)

REVIEW

Fishes

Batjakas et al. “Lionfish diet composition in in three study sites in the Aegean Sea (southern Crete, and Kastellorizo and Nysiros islands): An invasive generalist?”

General comments: The authors present the results of a descriptive study of lionfish gut contents from samples contributed by fishermen at three sites in the Aegean Sea. A variety of statistical approaches are used to show differences and similarities in stomach content composition among the three sites, and these analyses are used to discuss lionfish feeding behavior with respect to generalist vs. specialist strategies. While I thought the paper was adequate as a simple description of the gut contents differences, I thought the lack of characterization of potential differences in prey availability and habitat type among the three sites weakened the authors’ inferences about dietary generalist vs. specialist behavior. There were also a number of typographical errors and figures and tables that were formatted in difficult to interpret ways and lacking clear explanation in the figure and table legends. A lack of explanation in sections of the methods also led to ambiguity that needs to be resolved. I think the paper could be reconsidered after revision, but is not suitable for publication in its current state.

Line by line comments:

17-19: It’s unclear what sampling unit the low and high ends of the percentage ranges are based on. Are these maximum and minimums per individual fish sampled? Maximum and minimum site means?

21-25: How lionfish can be considered both generalists and specialists, in different context or scales, is a complex idea that is not explained as clearly as it could be here. This concept seems to be a crux of the hypothesis and interpretation of the study, and should be developed better in the introduction, as well.

31: Of the ~1000 documented invasive species, how many are fishes, and what are the primary vectors of invasion?

34: It is unclear what “short migration rate” means. Is this referring to rapid range expansion of invading fish populations?

41: It would be helpful to state the year of the first appearance of lionfish in the Mediterranean and briefly review the invasion history there in comparison with the invasion history in the Western Atlantic. Important insights into how environmental conditions, native prey communities, and management styles affect the invasion could be gained by consideration of the results of this study in relation to results of published studies from the Western Atlantic. In general, aiming for greater, global insights into the lionfish species and invasion ecology in general would improve the paper.

66: It is implied that the fish were provided by local fishermen. Do you have any information on where and how they were collected, i.e. by hook and line, nets, or spearing, and at approximately what depths and in what kinds of habitats they were collected? Were they caught as bycatch or through targeted efforts by the fishermen? As much information as possible about the environments in which the fish were collected should be presented, even if it is qualitative information.

68: 0.01 mm is an exceedingly high precision that seems unrealistic and unnecessary for TL measurements of adult fish. Is this a typographical error?

76: How were the fish preserved until processing- frozen?

109: Was there a wide range in size among prey species such that %F and %N might both exaggerate the importance of small but numerous species in the diet of lionfish? If so is there any way you might calculate a sort of prey weight-adjusted importance index? This potential caveat of interpretation should be discussed, at least.

133-137: It is unclear what aspects of the pi vs %F plot provide information about feeding strategy and trophic niche width. A brief review of how this biplot method works, and what patterns would be expected for different feeding strategies, would be helpful.

164: The legend for Figure 2 needs to explain what the green clouds are. I assume they are some kind of minimum and maximum values from resampling with randomization of stomach order.

180-190: The semi-hierachical classification of species and groups in the stomach contents table is confusing, especially because there is some overlap in categories. E.g., “benthic invertebrates” includes some crustaceans. The table legend includes some typographical errors and non-English letters.

189: Would it be possible to put confidence intervals or other error bars on the %F values in Figure 3? The figure legend also needs a little more description and has some typographical errors.

203: Figure 4 is helpful, but the small, dark colored fonts of the sub-taxon labels are difficult to read.

225: Does each marker in figure 5 represent an individual fish stomach analyzed? It seems like there should be more markers, especially for Nisyros.

311: Figure 6 seems to be missing Kastellorizo.

Author Response

thank you for your valuable comments and suggestions. we tried to respond giving the necessary clarifications

Reviewer 2 Report

when was the first detection of this species in the Meditarenian?

what is the actual abundance?

always indicate wheteryou are speaking of a native or an invasive population

In the conclusion part you should point out your findings and you should not refere to results of other studies

the first sentences of the introduction are hard to read

Author Response

thank you for your valuable comments and suggestions. we tried to respond and give the necessary clarifications

Round 2

Reviewer 1 Report

A few key changes have been made that fix errors and clarify the methods and findings of this study, and some helpful, additional background information is included. I think the changes are sufficient that this now merits publication in Fishes. 

There are still some minor English grammar and formatting problems, such as lack of subscripts and superscripts in mathematical formulae, and inconsistent capitalization and order of legend entries in the figures, but those can probably be fixed in the copy editing process. There is a discrepancy between the manuscript and the cover letter regarding the precision of TL measurement- one says 0.01 cm and the other says 0.1 cm. 

Author Response

Sorry about the discrepancy regarding the precision  of TL measurement. it is 0.1 cm and it is corrected.